# *iHeard* STL: Development and first year findings from a local surveillance and rapid response system for addressing COVID-19 and other health misinformation

**Kimberly J. Johnson**[1], **Olivia Weng**[1], **Hannah Kinzer**[1], **Ayokunle Olagoke**[2], **Balaji Golla**[1], **Caitlin O'Connell**[1], **Taylor Butler**[1], **Yoseph Worku**[1], **Matthew W. Kreuter**[1]*

**1** Brown School, Washington University in St. Louis, St. Louis, MO, United States of America, **2** School of Health and Kinesiology, University of Nebraska at Omaha, Omaha, NE, United States of America

* mkreuter@wustl.edu

**Data Availability Statement:** We have reviewed our consent form and consulted our Institutional Review Board. We have concluded that our

## Abstract

### Background

The U.S. Surgeon General and others have emphasized a critical need to address COVID-19 misinformation to protect public health. In St. Louis, MO, we created *iHeard STL*, a community-level misinformation surveillance and response system. This paper reports methods and findings from its first year of operation.

### Methods

We assembled a panel of over 200 community members who answered brief, weekly mobile phone surveys to share information they heard in the last seven days. Based on their responses, we prioritized misinformation threats. Weekly surveillance data, misinformation priorities, and accurate responses to each misinformation threat were shared on a public dashboard and sent to community organizations in weekly alerts. We used logistic regression to estimate odds ratios (ORs) for associations between panel member characteristics and misinformation exposure and belief.

### Results

In the first year, 214 panel members were enrolled. Weekly survey response rates were high (mean = 88.3% ± 6%). Exposure to a sample of COVID-19 misinformation items did not differ significantly by panel member age category or gender; however, African American panel members had significantly higher reported odds of exposure and belief/uncertain belief in some misinformation items (ORs from 3.4 to 17.1) compared to white panel members.

### Conclusions

Our first-year experience suggests that this systematic, community-based approach to assessing and addressing misinformation is feasible, sustainable, and a promising strategy

consent form does not allow us to share individual level de-identified data that is required for some of the analyses conducted in the manuscript. In addition, because of the small number of participants, the data cannot be fully anonymized at the individual level for sharing with 0% risk of identification. Aggregate data used for paper analyses is shared along with the code for all of the analyses that were conducted. For more information on sharing research data see "Sharing Research Data" in the WU Human Research Protections Research Guide https://online. fliphtml5.com/ikcz/ifub/#p=151. See https://hrpo. wustl.edu/about-us/contact-us/ for Institutional Review Board contact information for questions.

**Funding:** This work was supported by NIH/CEAL (1OT2HL161614-01). The funders had no role in study design, data collection and analysis, decision to publish, or preparation of the manuscript.

**Competing interests:** The authors have declared that no competing interests exist.

for responding to the threat of health misinformation. In addition, further studies are needed to understand whether structural factors such as medical mistrust underlie the observed racial differences in exposure and belief.

## Introduction

Misinformation is defined as "false or inaccurate information" [1]. During the COVID-19 pandemic, misinformation impacted vaccine uptake, promoted the use of treatments with unknown efficacy, and led to violence directed at workers, including healthcare, airline, and other front-line workers [2]. There is a need to systematically identify and respond to health misinformation to decrease its impact on individual and population health, especially within communities that have had higher rates of infection and mortality such as African American communities [3–5], to decrease its impact on individual and population health.

Public health surveillance is an essential tool in disease prevention efforts and for controlling morbidity and mortality. Surveillance systems track and report cases, identify and assess emerging threats, and help inform public health strategies aiming to protect and improve the health of populations [3]. Misinformation is increasingly recognized as a threat to the public's health, and several approaches to misinformation surveillance and response are being explored [4–7]. Each approach has limitations, including infrequent data collection, reliance exclusively on selected data sources such as social media conversations, and lack of a rapid-response mechanism and distribution infrastructure to disseminate findings and accurate information to local communities where the misinformation is circulating. As one example, in a January 2022 review of COVID-19 misinformation surveillance and response by 50 U.S. state health departments, only 34% of states had content on their websites addressing misinformation, and the most recent update, if noted, was dated three months prior [8].

In this paper, we describe the development of and initial findings from the first year of implementing a community-level misinformation surveillance and rapid response system. Using data and examples, we illustrate how the system tracks, reports, and responds to misinformation, what has been learned about misinformation trends over time, and which subgroups appear most vulnerable to misinformation.

## Methods

This study was approved by the Washington University in St. Louis Institutional Review Board.

### Background

In the early stages of the pandemic, frontline workers in St. Louis reported frequent exposure to misinformation from community members, and frustration that they felt unprepared to respond to it effectively and in the moment. *iHeard STL* (https://hcrlweb.wustl.edu/iheardstl/) was developed at Washington University in St. Louis by the Health Communication Research Lab (https://hcrl.wustl.edu/) to help solve this problem by proactively identifying misinformation circulating in the community and providing rapid, accurate responses to community organizations and the public to help them anticipate and counter misinformation. Weekly data collection is ongoing with funding support and this report describes the first year of data collection. Because *iHeardSTL* emerged as a local response system and not as a planned research project, sample size was based on feasibility issues rather than statistical power needed

to answer specific research questions. *iHeard STL* consists of three components: *surveillance*, *prioritization*, and *response*.

## Surveillance

**Panel members.**    We recruited panel members to answer brief, weekly mobile phone surveys assessing information they may have heard about COVID-19. We intentionally recruited from areas of St. Louis City and County in Missouri with a higher proportion of Black or African American residents and sought a 50/50 mix of community members and front-line workers. Front-line workers were defined on the recruitment survey as those who "...*regularly interact with members of the community for work in-person, online or over the phone. This can include healthcare workers, phone operators, or social service workers, for example.*" Outreach to potential panel members occurred primarily through community partners such as the St. Louis COVID-19 Regional Response Team (a collaborative of 35+ member organizations), St. Louis City Department of Health, St. Louis County Department of Public Health, YMCA of Greater St. Louis, United Way of Greater St. Louis, Alpha Phi Alpha Fraternity Inc., Herbert Hoover Boys and Girls Club and St. Louis Story Stitchers. We also recruited through the Washington University School of Medicine's Volunteer for Health Research Participant Registry [9], whose administrators sent an email to members who were ≥18 years old, Black or African American, and residing in zip codes under-represented in our sample.

We shared recruitment materials with our partners and/or distributed them while at community-based events hosted by our partners. Individuals could access the recruitment form through scanning a QR code on the recruitment flyer or email the study email address that was provided on the recruitment material. The recruitment form includes eligibility questions, followed by a full description of the project and what is asked of participants if they volunteer to participate. For those who wish to participate after reading the informed consent information, they provide contact information for payment purposes and to receive the longitudinal survey. Eligibility required being age 18 years or older, a resident of St. Louis City or St. Louis County, Missouri (or a non-resident employed there), and having access to a mobile phone with internet capabilities. Eligible individuals who provided informed consent and completed a brief baseline survey became panel members.

**Data collection.**    English language baseline and weekly surveys were designed in Qualtrics and optimized for mobile phone use. The first baseline surveys were completed 8/23/2021 and the first weekly surveys began less than one week later on 8/29/2021. Virus circulation in Missouri during this time as measured by new hospital admissions varied widely during the study period [10]. Masking policies during this time included mandates to strong recommendations for masking [11]. Enrollment and baseline surveys continued through the study period of 8/23/2021 to 8/21/2022, with the panel growing in size each week. Every Sunday, a link to the weekly survey was texted to all panel members. The survey closed 48 hours later. Panel members received a $10 Forte cash card for completing the baseline survey; $5 was added to the card electronically each time they completed a weekly survey.

**Measures.**    The baseline survey helped characterize panel membership and allowed us to stratify weekly misinformation findings by sub-groups within the panel. It assessed items including panel members' age, gender (male, female, non-binary/third gender, prefer not to say), race (white, Black or African American, American Indian or Alaska Native, Asian or Asian American, Native Hawaiian or Pacific Islander, other, prefer not to say), Hispanic ethnicity (yes/no/prefer not to say), COVID-19 vaccination status (both doses of a 2-dose vaccine/first dose of a 2-dose vaccine/1-dose vaccine/unvaccinated but plan to get a COVID-19 vaccine/unvaccinated and don't plan to get a COVID-19 vaccine/don't know), level of worry

**Table 1. Exposure and belief questions and responses.**

| Misinformation item | Exposure[a] | Belief[b] |
|---|---|---|
| VaxFail | The COVID-19 vaccines are failing. | When you heard, read or saw it, what was your first reaction? |
| VaxDanger | The COVID-19 vaccine is more dangerous than the virus itself. | |
| KidMask | Masks are dangerous for kids. | |
| Ivermectin | Taking Ivermectin will prevent or cure COVID-19, so vaccines are not needed. | |

[a]All exposure questions were prefaced with "In the last week, have you heard, read or seen something like this:" Possible responses were "Yes", "No", and "Not sure".

[b]If a panel member responded "Yes" to the exposure question; they were asked the belief question that was the same for all misinformation items. Possible responses were "Definitely True", "Seems like it could be true", "Not sure if it's true or untrue", "Seems like it's not true", and "Definitely not true".

about COVID-19 (0–100), and whether they or a loved one had ever had COVID-19 or been hospitalized with COVID-19 (yes/no/not sure).

The weekly surveys assessed exposure to, sources of, and belief in specific instances of COVID-19 misinformation. Panel members were first asked, "In the last 7 days, have you heard, read or seen <misinformation item>?" In the first 12 months of the project, 45 different items were included on at least one survey. Sample misinformation items include "*taking Ivermectin will prevent or cure COVID-19, so vaccines are not needed*" and "*the COVID-19 vaccine is more dangerous than the virus itself*." The exact wording for four misinformation items analyzed in this paper is shown in **Table 1**.

Selection of misinformation items were informed by two main sources: COVID-19 misinformation tracking websites (e.g., CDC's State of Vaccine Confidence Insights Reports [12], Public Health Communication Collaborative's Misinformation Alerts [13], Google FactCheck [14]) and an open-ended question on the weekly survey asking panel members what other information about COVID-19 they had heard in the last seven days.

On average, misinformation items were included in the survey for about seven consecutive weeks (range 1–23 weeks); we removed items when <15% of panel members reported hearing the item for at least three consecutive weeks, or items were not applicable anymore. For example, a misinformation item claiming that "COVID is caused by snake venom" was introduced for only one week because exposure was near zero among our panel members. In any given week, five to ten misinformation items were assessed. After each misinformation item, regardless of the panel member's response, the survey screen displayed accurate information about the item. This is a recommended best practice [15] to avoid introducing, validating, or spreading misinformation just by asking about it.

For panel members who reported that they had heard, read, or seen a misinformation item in the last seven days, the next question assessed where they heard or read or saw it (with responses of family member or friend/neighbor or coworker/someone else/on social media/ other internet source/TV or radio/other (please specify)/not sure/refuse to answer). Panel members who reported that they heard the misinformation item were also asked whether they believed it (with responses of definitely true/seems like it could be true/not sure if it's true or untrue/seems like it's not true/definitely not true). A final open-ended item asked them to share any other COVID-19 information they had heard in the last seven days.

Each Tuesday, when the weekly survey closed, panel member responses were aggregated and rapidly analyzed for use in summary reports to community partners and the *iHeard STL*

dashboard. We calculated *weekly exposure* to each item as the number of respondents who reported having heard/read/seen the item (with responses of Yes) in the last seven days among those who responded to the question. *Weekly belief* was calculated as the percentage of exposed respondents in a given week who reported that the item was definitely true/seems like it could be true/not sure if it's true or untrue. We also created *first exposure* and *first belief* variables for use in sub-group analyses to capture panel members' responses to these questions the first time they answered them, thereby reducing potential bias from the accurate information that was presented following each misinformation item on each survey. Panel members were categorized as being *first exposed* if they reported having heard/read/seen an item (with responses of Yes) the first time they answered a survey question about that item. Panel members were categorized as *first believing* if they reported any of three responses (definitely true/seems like it could be true/not sure if it's true or untrue) the first time they answered the belief question for a recurring misinformation item.

### Prioritization

We prioritized misinformation items according to threat level, which was calculated each week for each misinformation item as the product of the percent exposed and the percent *first belief* divided by 10,000 (the maximum possible product), generating a score between zero and 100. Threat score categories of low, medium, and high corresponded to scores of 0 to <5, ≥5 to <20, and ≥20, respectively. Threat level was used to identify misinformation items to highlight in our response.

### Response

Surveillance findings and responses to misinformation are rapidly reported back to the community in two ways. First, findings from each weekly survey, including threat-level priorities and exposure trend data over time, are posted to a public-facing dashboard [16] within 72 hours of the survey closing. For each misinformation item assessed that week, the dashboard provides accurate responses in a short, and longer version. The short version models a quick response that could be used if someone encountered the misinformation. The longer version provides a more detailed explanation, with evidence and hyperlinks to official and original sources, for users who want or need to know more.

Second, we develop and e-mail a weekly "alert" to community partners that identifies and describes a high priority finding from that week. Alerts include suggested action steps, and a link to the dashboard. Alerts were distributed through GoDaddy [17] from 3/24/2022 to 6/30/2022; we used its "unique view" tracker to determine how many and which partners opened our alert emails (referred to as a "view"), and its "engage" tracker to determine the number of alert recipients that clicked at least one of the links in the alert or shared our email with others [18]. If a subscriber opened the email multiple times only one view was counted.

### Data analyses

To illustrate aspects of the system, we conducted four sets of analyses examining: (1) time trends for *weekly exposure* to and *weekly belief* of misinformation; (2) variability in misinformation *first exposure* and *first belief* by sub-groups of panel members; (3) emerging misinformation identified from open-ended responses; and (4) use of the misinformation dashboard and weekly alerts. All analyses were conducted using R software.

**Time trends for weekly exposure to and weekly belief of misinformation.** We selected four misinformation items assessed in weekly surveys *a priori* to use as examples in analyses (**Table 1**). We plotted the proportion of panel members exposed to and believing (see

**S1 Table** for definitions) each misinformation item week-by-week. We used linear regression to estimate slopes (β's) and 95% confidence intervals (CIs) to estimate the percent change in exposure and belief over time as a function of the survey week number. We removed data points before 10/17/2021 due to the small number of enrolled panel members ($\leq 12$) in the first seven weeks of surveillance.

**Misinformation first exposure and first belief by sub-group.**   Using logistic regression, we examined associations between panel members' age category, race, gender, occupation category, and their exposure to and belief of misinformation. Separate models examined *first exposure* and *first belief* as outcomes (see **S1 Table** for definitions). Among the demographic predictors in each model, stratified analyses were performed by age category ($18$-$39/40$-$49/\geq$ $50$ years), race (white/Black or African American), gender (male/female), and occupation category (front-line worker/not). We excluded those with race and gender other than the categories listed above due to small numbers. The likelihood ratio (LR) test was used to compare models to assess whether a variable significantly improved model fit for *first exposure* or *first belief*. Results were considered statistically significant if the two-sided p-value was $< .05$. Odds ratios (ORs) and 95% confidence intervals (CIs) are reported.

**Identifying new misinformation from open-ended responses.**   To examine the potential for weekly misinformation surveillance to identify emerging misinformation, we coded open-ended responses provided by panel members when asked what else they had heard about COVID-19 in the last seven days. Because any participant's open-ended response could include multiple distinct claims (e.g., "vaccines are not effective and masking hurts children"), the unit of analysis for coding was each unique claim reported in a single week.

Claims were assessed for containing misinformation if they could be proven or disproven with peer-reviewed scientific articles, public data, public reports, or news articles from fact-checked news outlets (e.g., New York Times, Washington Post). For this reason, claims such as, "nothing to report," and "My graduation ceremony required masks" were not assessed for misinformation. Two research team members independently coded each response into one of six exclusive categories (vaccine, disease, virus, healthcare, general, not myth) and up to two of 47 non-exclusive sub-categories (see **S2 Table**). The codebook was developed using an inductive, iterative approach [19,20]. Claims were classified as misinformation when they were false or unproven given best available evidence at the time [2]. Discrepancies between coders were resolved through discussion. We report the frequency of misinformation claims overall and by category, and the proportion of panel members providing open-ended responses.

**Use of misinformation dashboard and alerts.**   Using web analytics, we report total page views of the *iHeard STL* dashboard from 3/24/2022 to 9/30/2022. We also report use of weekly misinformation alerts in terms of opened e-mails and clicks to access the *iHeard STL* dashboard from 3/24/2022 to 6/30/2022.

## Results

### Participants and response rates

Between 8/23/2021 and 8/21/2022, we enrolled 214 panel members. Most were female (63.1%) and either white (52.3%) or Black/African American (37.4%) (**Table 2**); the mean age was $38.6 \pm 14.6$ years (not shown). Over half were front-line workers (54.7%) and most reported at baseline being fully vaccinated with a two-dose vaccine (85.0%) (**Table 2**). Response rates for each weekly survey ranged from 72.5% to 100% with a mean of $88.8\% \pm 4.8\%$ (**Fig 1**). In all, 7,512 surveys were sent and 6,605 were completed.

**Table 2. Panel member characteristics.**

|  | Overall (N = 214) |
|---|---|
| **Gender** |  |
| Male | 75 (35.0%) |
| Female | 135 (63.1%) |
| Non-binary / third gender | 3 (1.4%) |
| Prefer not to say | 1 (0.5%) |
| **Race** |  |
| white | 112 (52.3%) |
| Black or African American | 80 (37.4%) |
| Asian or Asian American | 6 (2.8%) |
| American Indian or Alaska Native | 0 (0%) |
| Multiple races | 7 (3.3%) |
| Other | 3 (1.4%) |
| Prefer not to say | 6 (2.8%) |
| **Hispanic Ethnicity** |  |
| Yes | 16 (7.5%) |
| No | 196 (91.6%) |
| Prefer not to say | 2 (0.9%) |
| **Age Category (years)** |  |
| 18–29 | 70 (32.7%) |
| 30–39 | 61 (28.5%) |
| 40–49 | 30 (14.0%) |
| 50–59 | 23 (10.7%) |
| ≥60 | 30 (14.0%) |
| **Vaccination Status** |  |
| Yes, both doses of a 2-dose vaccine | 182 (85.0%) |
| Yes, first dose of a 2-dose vaccine | 1 (0.5%) |
| Yes, a 1-dose vaccine | 13 (6.1%) |
| No, but I plan to get a COVID-19 vaccine | 6 (2.8%) |
| No, and I don't plan to get a COVID-19 vaccine | 7 (3.3%) |
| Prefer not to answer | 5 (2.3%) |
| **Participant Type** |  |
| Community Member | 96 (44.9%) |
| Front-line Worker | 118 (55.1%) |

## Time trends in weekly exposure and weekly belief

Weekly exposure rates for the four misinformation items showed significant declines over time. Average per week decreases were -8.1% (95% CI: -10.2 to -6.0) for KidMask, -3.3% (95% CI: -5.4 to -1.3) for VaxFail, -2.7% (95% CI: -3.9 to -1.5) for VaxDanger, and -2.6% (95% CI: -4.0 to -1.2) for Ivermectin. The rate of weekly belief did not significantly change over time for any of the four misinformation items (**Fig 2**, **S3 Table**).

## Misinformation first exposure and first belief by sub-group

*First exposure* did not differ significantly by panel member age category, gender, or occupation category for any of the four example misinformation items (LR test $p > .05$) (**Table 3**). However, Black or African American panel members were more likely than white panel members to be exposed to the VaxDanger item (OR = 3.4; 95% CI: 1.4–8.2; $p_{\mathrm{VaxDanger}} = .006$), but not other misinformation items ($p_{\mathrm{VaxFail}} = .32$, $p_{\mathrm{KidMask}} = .61$, $p_{\mathrm{Ivermectin}} = .34$).

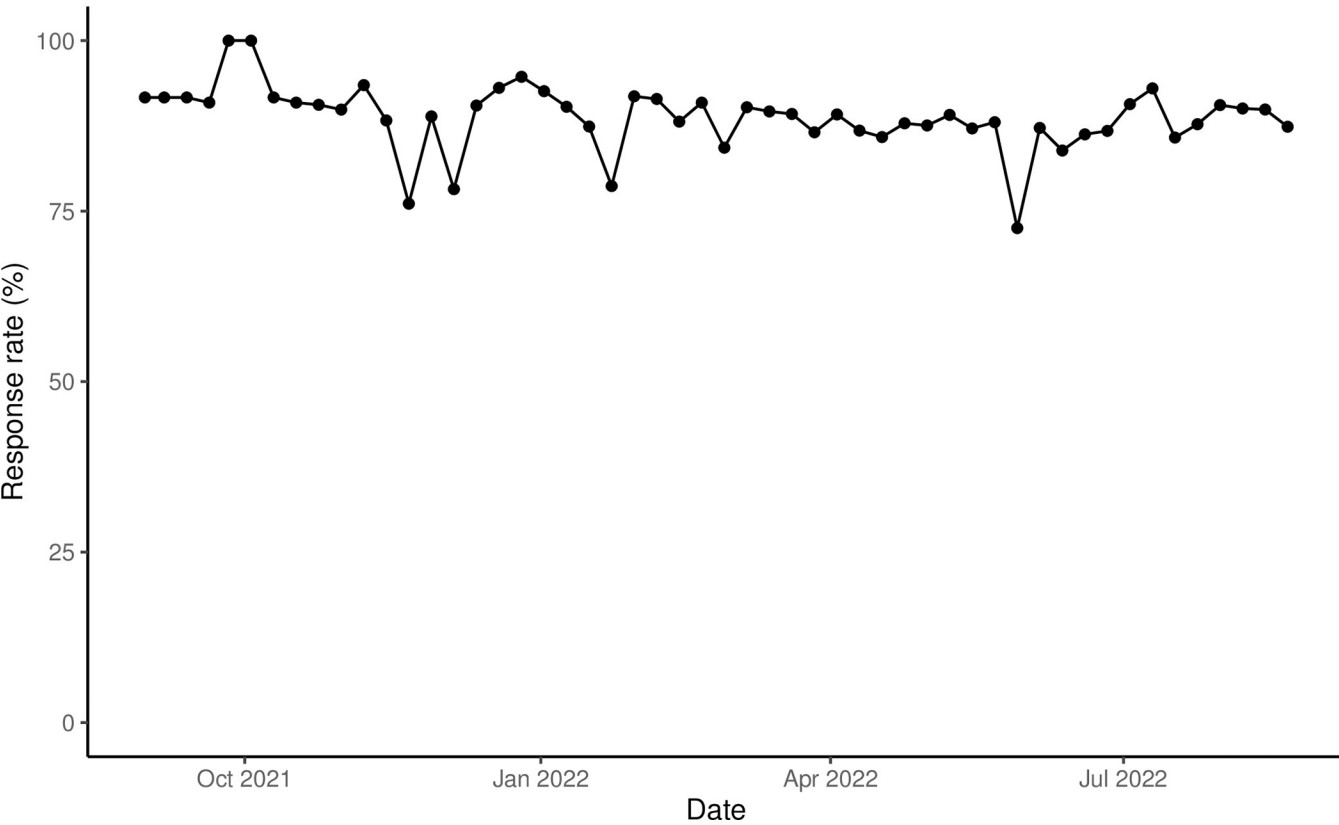

**Fig 1. The weekly response (%) from August 2021 to August 2022.**

*First belief* of misinformation was significantly higher for Black or African American vs. white panel members; odds ratios for believing the VaxDanger, VaxFail, and Ivermectin items were, respectively, 17.1 (95% CI: 2.6–339.1), 6.3 (95% CI: 1.6–28.4), and 5.3 (95% CI: 1.2–25.5). Panel members ages 50 and older had an 8.4 (95% CI: 1.5–67.5) times higher odds of believing the KidMask item compared to those 18 to 39.

## Open-ended response analysis

Across 52 weekly surveys, 165 different panel members (77% of total enrolled) provided 1,449 open-ended responses about other COVID-19 information they had heard in the last week. Of these, 1,136 were assessable claims. Among all claims, the most frequent classification for main category was "vaccine" (488, 43%), and the most frequent subcategory was "statistics" (250, 13%). Among misinformation claims, the most frequent main category was also "vaccine" (257, 52%), and the most frequent subcategory was "booster" (78, 9%) (**Table 4**).

On average, about 19 panel members (18.71 ± 8.37) provided at least one assessable claim per week, for an average of 0.18 ± 0.12 claims per respondent per week. Of the 1,136 assessable claims, 505 (44%) were classified as misinformation, with 112 panel members ever providing at least one open-ended response that was classified as misinformation. The average number of respondents per week who reported at least one misinformation claim was 9 (8.69 ± 5.57). Overall, there was an average of 0.09 ± 0.08 misinformation claims per respondent per week.

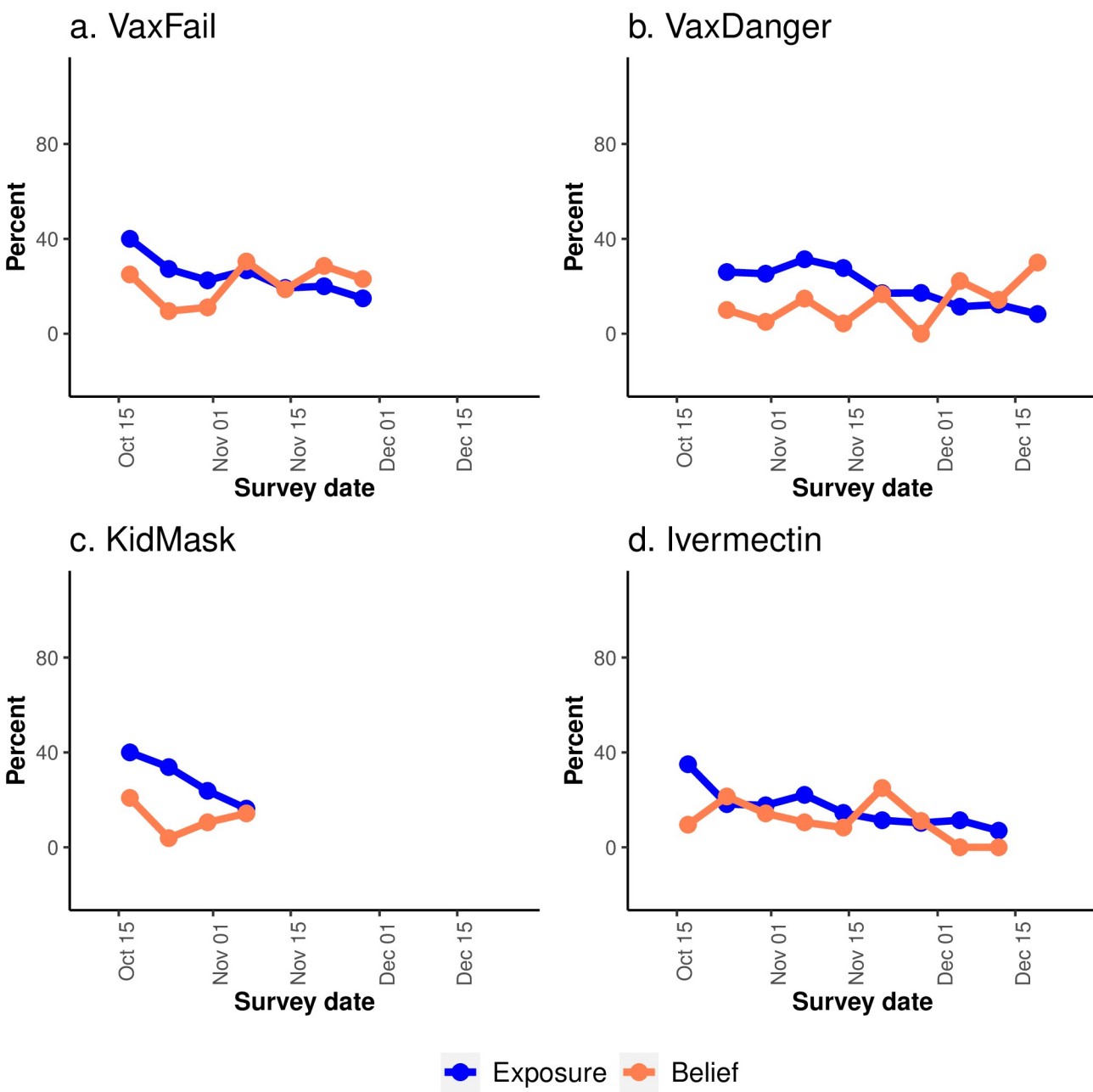

**Fig 2.** Trends of weekly exposure (blue) and weekly belief (coral) of four inaccurate information items: (a) VaxFail; (b) VaxDanger; (c) KidMask; (d) Ivermectin.

### Use of misinformation dashboard and alerts

The *iHeard STL* dashboard was promoted by email starting 3/24/2022. Through 9/30/2022, we recorded 15,939 views of the homepage, which provides information about exposure to specific misinformation threats in the last week, beliefs in the misinformation, time trends for misinformation exposure, and a short, accurate response to address each misinformation item. Of the 15,939 views, which included Washington University internal users, 611 (4%) then clicked to receive "more information" about a misinformation threat, which took them to a

**Table 3. Odds ratios of first exposure and belief by panel member demographic characteristics[a].**

| Variable | First exposure | | | | First belief | | | |
|---|---|---|---|---|---|---|---|---|
| | VaxFail | VaxDanger | KidMask | Ivermectin | VaxFail | VaxDanger | KidMask | Ivermectin |
| | OR (95%CI) | OR (95%CI) | OR (95%CI) | OR (95%CI) | OR (95%CI) | OR (95%CI) | OR (95%CI) | OR (95%CI) |
| **Age** | | | | | | | | |
| 18–39 | 1.0 (ref) | 1.0 (ref) | 1.0 (ref) | 1.0 (ref) | 1.0 (ref) | 1.0 (ref) | 1.0 (ref) | 1.0 (ref) |
| 40–49 | 0.68 (0.21–2.04) | 1.20 (0.35–3.61) | 1.57 (0.51–4.91) | 1.68 (0.56–4.83) | 1.25 (0.16–6.75) | 0.90 (0.04–6.99) | 1.50 (0.07–17.57) | NA |
| 50 and older | 0.86 (0.31–2.32) | 2.11 (0.81–5.38) | 2.10 (0.76–5.97) | 1.58 (0.59–4.10) | 2.00 (0.44–8.55) | 1.64 (0.21–9.64) | **8.44 (1.51–67.47)** | 2.56 (0.54–11.62) |
| **Gender** | | | | | | | | |
| Male | 1.0 (ref) | 1.0 (ref) | 1.0 (ref) | 1.0 (ref) | 1.0 (ref) | 1.0 (ref) | 1.0 (ref) | 1.0 (ref) |
| Female | 0.52 (0.20–1.37) | 0.56 (0.23–1.41) | 0.94 (0.36–2.52) | 0.72 (0.30–1.81) | 1.57 (0.34–11.24) | 1.81 (0.27–35.90) | 0.40 (.08–2.29) | 0.46 (0.10–2.50) |
| **Race** | | | | | | | | |
| white | 1.0 (ref) | 1.0 (ref) | 1.0 (ref) | 1.0 (ref) | 1.0 (ref) | 1.0 (ref) | 1.0 (ref) | 1.0 (ref) |
| Black or African American | 1.60 (0.64–4.04) | **3.40 (1.43–8.21)** | 0.78 (0.29–2.03) | 0.64 (0.24–1.55) | **6.30 (1.59–28.44)** | **17.14 (2.62–339.14)** | 3.78 (0.61–23.70) | **5.28 (1.18–25.46)** |
| **Participant Type** | | | | | | | | |
| Community Member | 1.0 (ref) | 1.0 (ref) | 1.0 (ref) | 1.0 (ref) | 1.0 (ref) | 1.0 (ref) | 1.0 (ref) | 1.0 (ref) |
| Front-line worker | 1.34 (0.54–3.44) | 1.42 (0.63–3.34) | 0.54 (0.21–1.37) | 1.11 (0.51–2.49) | 2.06 (0.46–14.52) | 2.77 (0.43–54.34) | 0.38 (0.08–1.81) | 1.50 (0.32–10.86) |

[a]**S4 Table** provides cross tabulated counts of first exposure and first belief by subgroup after excluding those who selected a non-binary/third gender or "other" gender category for gender analyses and those who did not report race or selected a race other than Black/African American or white for race analyses.

longer response with a more detailed explanation, evidence, and citations to support the accurate response.

We began sending weekly alerts by email on 3/24/2022. Until 6/30/2022, the number of non-university community organizations alert subscribers ranged from 36 to 46 per week (mean per week = 42.5 ± 3.3). The weekly view rate for alerts ranged from 22.7% to 51.4%, with a mean of 32.5% ± 8.9% (**Fig 3**). The weekly engagement rate–the proportion of those viewing the alert who then clicked on a hyperlink to take them to the dashboard–ranged from 0 to 31.3% by week, with a mean of 13.3% ± 10.2%.

**Table 4. Misinformation claims by main categories and top 5 sub-categories.**

| Category | N (% of 505 total misinformation claims) | Example |
|---|---|---|
| **Top 5 main categories** | | |
| Vaccine | 257 (52%) | "That older people—including Betty White—might be dying from the booster shot"– 2022-1-16 |
| General | 162 (33%) | "Masks dont help"– 2022-5-222 |
| Disease | 50 (10%) | "Omicron cases do not require hospitalization"– 2022-1-23 |
| Virus | 22 (4%) | "This is man made and will continue to vary for the next two years"– 2021-12-12 |
| Healthcare | 7 (1%) | "There are major heathcare, first responder, and other essential worker shortages due to vaccine mandates."– 2021-10-31 |
| **Top 5 sub-categories** | | |
| Booster | 78 (9%) | "That older people—including Betty White—might be dying from the booster shot"– 2022-1-16 |
| Variants | 68 (8%) | "The vax is causing the variants."– 2021-12-05 |
| Development | 67 (8%) | "Covid-19 is over"– 2022-05-22 |
| Effectiveness | 66 (8%) | "Vaccinated infected at same rate as unvaxxed"– 2021-12-05 |
| Statistics | 63 (7%) | "Masks dont help"– 2022-5-22" |

[a]Main categories were mutually exclusive and each assessable claim was assigned a main category.

[b]Sub-categories were not mutually exclusive; a claim could have no sub-category, one sub-category, or two sub-categories.

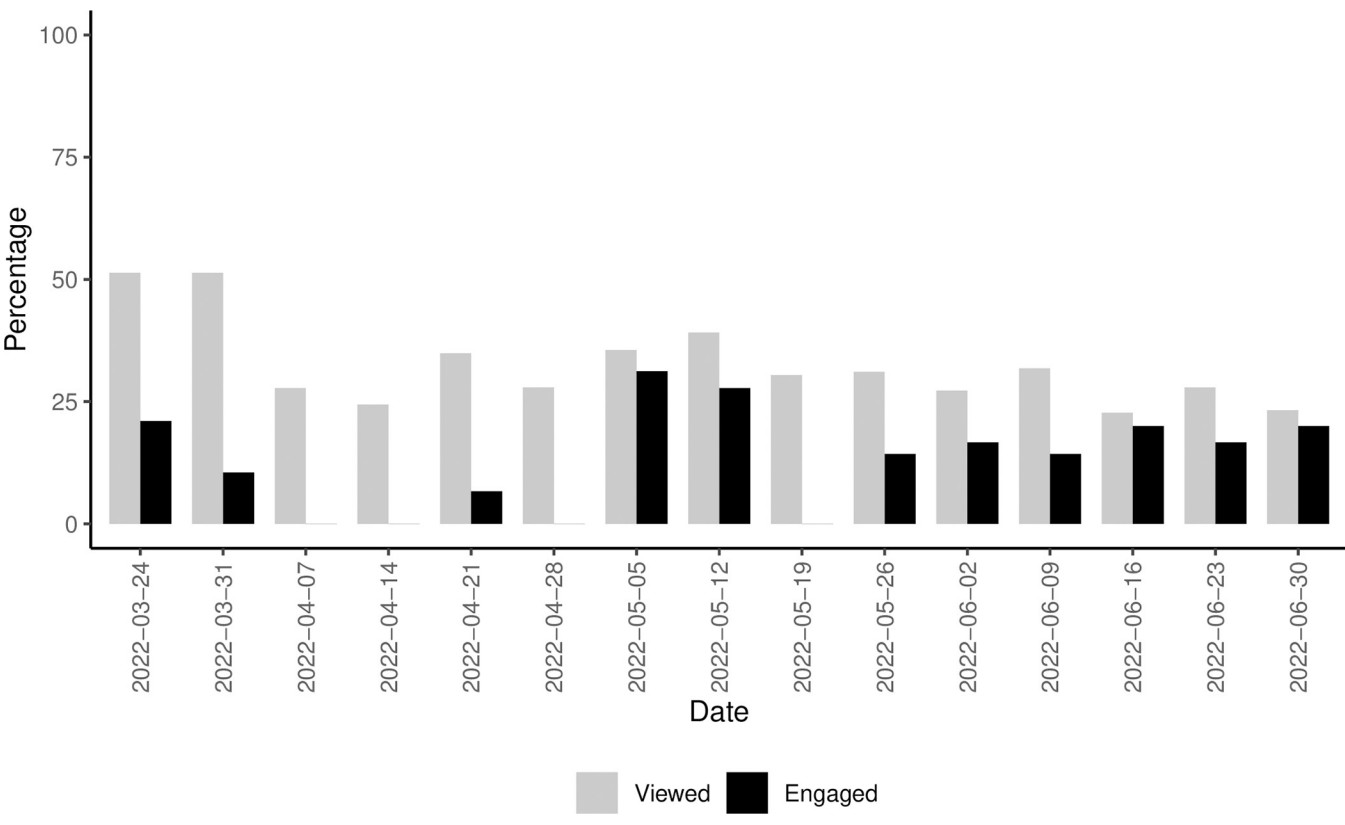

**Fig 3. Percent of non-Washington University email recipients who viewed (grey) and engaged (black) in weekly misinformation alerts from 3/24/2022 to 6/30/2022.**

## Discussion

Rapid, local responses to COVID-19 and other health misinformation are urgently needed [18]. We developed one of the first local misinformation surveillance and response systems in the United States. Our system routinely tracks and rapidly reports specific claims about COVID-19 that are circulating locally and captures new information that community members are hearing about other emerging health issues. Because these data are collected each week, we can identify near real-time trends, helping to prioritize community responses in the face of multiple misinformation threats, and even targeting those efforts to population subgroups with disproportionate exposure to and/or belief in specific misinformation.

We found that community exposure to misinformation items generally decreased over time, but the rate varied by misinformation topic. For example, estimated exposure to an item about "masks are dangerous for kids" had the sharpest estimated decrease over time at ~8.1% decline per week, while estimated exposure to an item about "the COVID-19 vaccine is more dangerous than the virus itself" decreased more slowly, at ~2.7% per week. Studies have identified a range of factors that contribute to the spread and duration of misinformation exposure, including "echo chambers" of like-minded clusters of individuals sharing a limited set of information channels; avoidance or selective non-exposure to fact-checked corrective evidence in a crowded media environment; and increasingly rapid news cycles [15,21–23]. Although our misinformation surveillance system alone cannot yet distinguish the relative contributions of different factors, the weekly collection of exposure and belief data at a community level makes

it possible to integrate data from other sources into analyses to provide important insights that could inform strategies to better address misinformation.

In contrast to declining exposure over time, initial findings from our system suggest that believability of misinformation claims was relatively stable among our panel members. These data reinforce prior research findings that misinformation beliefs are difficult to change over time [24–27]. The stability of belief estimates is particularly noteworthy because the weekly survey provided respondents with a brief counter-message immediately after panel members answered each question, each week, about exposure to a misinformation item. In other words, in line with previous literature [24–27], we found that belief in some misinformation claims persisted in the face of routine counter-messaging.

These findings highlight the potential of inoculating or "pre-bunking" community members against misinformation to prevent its spread [28]. By identifying misinformation claims with stable believability, counter messaging could be supplemented with theory-based interventions to address the sources of entrenched beliefs. For example, in Zimbabwe, a behavioral theory-informed intervention to address misinformation about HIV and condom usage successfully changed beliefs associated with intention to use condoms [29]. Providing trusted local messengers with accurate information to share with constituents could further enhance and accelerate efforts to address misinformation.

Black or African American panel members had greater odds of reported exposure to and belief in misinformation. Compared to white panel members, their odds of exposure to claims that the vaccine was more dangerous than the disease were three times greater, and they also had higher odds of believing an item could be true or being uncertain whether it's true or untrue for three out of four misinformation items. There is a long history of African Americans being targeted with health harming products and information as well as mistrust of health care and health research among Africans Americans based on past abuses [30,31]. For our project's purpose of helping to focus and support community responses to misinformation, the ability to identify specific sub-groups at increased risk of misinformation exposure or belief or uncertain belief is particularly valuable. For example, it could inform message and channel strategies such as outreach through specific community partners or trusted messengers with unique access and credibility in a given sub-group. It could also help guide efforts to better understand the factors underlying differences between sociodemographic groups [32].

The World Health Organization recommends monitoring publicly driven conversation online and offline about vaccine sentiment. Current approaches rely heavily on social media listening or occasional cross-sectional surveys [33]. Our weekly community surveillance efforts complement these approaches by detecting when specific misinformation threats enter a community and how they spread.

In our analyses of over 1,000 open-ended responses to a question about what COVID-19 information panel members had heard in the last 7 days, nearly half of the claims were classified as misinformation (45%). These claims were invaluable in shaping surveillance, as 21 of the 45 survey items we administered in the first year originated as open-ended responses. This illustrates a key benefit of routine misinformation surveillance: it feeds a rapid-cycle process that converts new misinformation reported by a few community members into a community-wide survey within days. If or when community-wide survey responses show that the new misinformation poses a substantial threat, we can immediately spread accurate information to counter it. This echoes other misinformation response initiatives that use community-engaged methods such as citizen science and participatory surveillance to rapidly identify health-relevant events and complement traditional surveillance system methods [34–36].

Community-engaged approaches to surveillance may help build trust between public health institutions and communities [36]. For example, it provides community members an

opportunity to voice concerns about misinformation and observe how local systems respond to their concerns. At the same time, health professionals can learn about misinformation that is circulating locally and focus their efforts accordingly. Establishing infrastructure that enables community members to be active participants in identifying misinformation locally may help foster community information stewardship [37].

## Strengths and limitations

*iHeard STL* is a novel, real-world, local surveillance and response system for health misinformation. Both the system and this initial evaluation have several strengths. First, they are multi-faceted. The system includes a weekly surveillance apparatus, a public-facing dashboard platform to share results with the community, and multiple mechanisms for rapidly distributing accurate information to community organizations to help them address misinformation when they encounter it. This report shares first-year findings from all components of the system. Second, community engagement with the system has been high. Response rates to weekly surveillance have averaged 88%, two-thirds of panel members have contributed open-ended responses to share new misinformation they have heard, and community organizations and residents are accessing the accurate information provided by the system.

There are important limitations of this evaluation, mostly attributable to *iHeard STL* being rapidly developed and launched as a community response system rather than a research project. For example, it currently relies on a convenience sample of panel members, which may affect the generalizability of surveillance results. Women and vaccinated individuals are over-represented in our panel, and we intentionally over-recruited Black and African American individuals because they have been disproportionately affected by COVID-19 [38] and have higher rates of vaccination hesitancy, perhaps due to greater exposure to misinformation [39]. Our panel also includes many front-line workers, for whom the system was designed to help. In addition, an estimated 9.2% of our panel members are employed in the healthcare sector (16.5% of these reported being frontline workers) vs. 14% of the U.S. population employed in this sector [40]. This may cause underestimated exposure rates on surveyed health-related claims. Because the community panel was accrued over time, early weeks of surveillance had a smaller number of respondents, which led to some data being excluded from trend analyses for misinformation exposure and belief. Estimates for differential exposure to and belief of misinformation between sub-groups may be imprecise due to small numbers and should be interpreted cautiously. We also excluded individuals who reported a gender other than male or female from the gender analysis and who reported a race/ethnicity other than white or African American from the race analysis, which limits the interpretation of these results to the included groups. In addition, the small sample size with low statistical power prohibited us from performing multivariable regression models. Finally, our surveys were conducted in English and therefore we may have missed capturing differences in misinformation and patterns circulating in non-English speaking communities.

## Conclusions

It is feasible to create and operate a community-based system to monitor and rapidly respond to health misinformation. After one year of operation, we find that the data captured through routine surveillance can help identify priority threats in a timely way, as well population sub-groups that may be disproportionately affected.

Our system has evolved a great deal in its first year and will continue to evolve in at least three specific ways: assessing and addressing non-COVID health misinformation (e.g., Mpox and other emerging diseases); assessing and reinforcing accurate health information (e.g.,

approval of boosters); and expansion to include other communities in other states. We will also examine longer-term and community level effects of the system, such as its impact on scientific trust, health knowledge, and related health improvement in communities.

## Supporting information

**S1 Table. Exposure and belief definitions, measures, and use.**
(DOCX)

**S2 Table. Subcategories for open-ended misinformation survey responses.**
(DOCX)

**S3 Table. Beta estimates for first exposure and first belief rate.**
(DOCX)

**S4 Table. Crosstab of raw counts of first exposure and first belief by sub-group.**
(DOCX)

**S1 File.**
(ZIP)

## Author Contributions

**Conceptualization:** Matthew W. Kreuter.

**Data curation:** Balaji Golla.

**Formal analysis:** Kimberly J. Johnson, Olivia Weng, Hannah Kinzer, Ayokunle Olagoke, Caitlin O'Connell, Yoseph Worku.

**Methodology:** Kimberly J. Johnson, Ayokunle Olagoke, Matthew W. Kreuter.

**Project administration:** Ayokunle Olagoke, Taylor Butler, Matthew W. Kreuter.

**Supervision:** Kimberly J. Johnson, Matthew W. Kreuter.

**Visualization:** Olivia Weng, Ayokunle Olagoke, Balaji Golla, Yoseph Worku.

**Writing – original draft:** Kimberly J. Johnson, Olivia Weng, Hannah Kinzer, Ayokunle Olagoke, Taylor Butler.

**Writing – review & editing:** Kimberly J. Johnson, Olivia Weng, Hannah Kinzer, Ayokunle Olagoke, Caitlin O'Connell, Matthew W. Kreuter.

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
