## [Decision Letter · Decision Letter 0]

25 Jul 2023

PONE-D-22-32748*iHeard* STL: Development and first year findings from a local surveillance and rapid response system for addressing COVID-19 and other health misinformationPLOS ONE

Dear Dr. Kreuter,

Thank you for submitting your manuscript to PLOS ONE. After careful consideration, we feel that it has merit but does not fully meet PLOS ONE’s publication criteria as it currently stands. Therefore, we invite you to submit a revised version of the manuscript that addresses the points raised during the review process.

We look forward to receiving your revised manuscript.

Kind regards,

Jerome Nyhalah Dinga, PhD

Academic Editor

PLOS ONE

Journal Requirements:

a) Did participants provide their written or verbal informed consent to participate in this study?

"Funding 

 This work was supported by NIH/CEAL (1OT2HL161614-01)."

"This work was supported by NIH/CEAL (1OT2HL161614-01). The funders had no role in study design, data collection and analysis, decision to publish, or preparation of the manuscript."

Reviewers' comments:

Reviewer's Responses to Questions

**Comments to the Author**

1. Is the manuscript technically sound, and do the data support the conclusions?

Reviewer #1: Yes

Reviewer #2: Yes

2. Has the statistical analysis been performed appropriately and rigorously? 

Reviewer #1: Yes

Reviewer #2: Yes

3. Have the authors made all data underlying the findings in their manuscript fully available?

Reviewer #1: Yes

Reviewer #2: Yes

4. Is the manuscript presented in an intelligible fashion and written in standard English?

Reviewer #1: Yes

Reviewer #2: Yes

5. Review Comments to the Author

Reviewer #1: General comments - This paper addresses an important topic of misinformation during pandemic times, when members of the public were exposed to high volumes of accurate information, misinformation, and disinformation. The study employed a public health surveillance model to assess prevalence of misinformation, and had high survey response rates. The study generated very interesting findings including the feasibility of a local misinformation surveillance system, and the persistence of misinformation beliefs over time despite routine counter-messaging.

Introduction

- Consider including a definition of “misinformation.”

- Consider including more information around COVID disparities to set the stage for why you specifically recruited a large proportion of Black community members to be a part of the panel.

Methods

- Panel composition. The authors describe their efforts to recruit a diverse panel with intentionality to include members of certain under-represented demographics including Black community members and community members from certain zip codes. They also sought representation of front-line workers in the sample and provide a definition they used for this term. It is worth noting that front-line workers encompassed healthcare workers – are these healthcare workers representative of the general population when it comes to receiving health information? Among the front-line workers, do the authors know what proportion were in healthcare?

- The authors should include information in the methods regarding timing of their surveys in relation to the pandemic. Was this conducted at a time of surge or low virus circulation? What were local public policies at the time? (I believe their surveys started around the time or shortly after the Delta surge in the summer of 2021).

- The authors provide a thorough description of how beliefs and exposures were captured.

- Were surveys provided in English only? If so, would this help explain the low participation of Hispanic community members? Language of surveys should be noted in methods, and potential biases should be noted in limitations.

Discussion

- For the paragraph starting in line 383, nuanced framing of discussion in differences between racial/ethnic groups is warranted. This is especially important given stark inequities in COVID-related health outcomes between these groups. Readers should not walk away from this paragraph thinking that members of the Black community are more “gullible” (ie likely to believe false information) due to their “culture.” This is somewhat implied as the paragraph currently stands, though I do not think this was the authors’ intent. This paragraph must include a more robust discussion,including explicit consideration of how structural factors including structural violence, institutional racism, and even interpersonal racism influence trust in biomedical health system and where community members access health information. It is very important to specifically call out racism and its impact on mistrust of the biomedical system. See for example: https://www.ncbi.nlm.nih.gov/pmc/articles/PMC7241063/

- (Similarly, the framing of racial differences in the abstract must also be addressed. The current wording regarding differences in belief of misinformation between Black and white community members is not nuanced enough and does not adequately address the systemic factors. )

- The exclusion of members from certain demographics in the analysis due to small numbers may have biased the data. This should be noted in the limitations.

- The limitations can also address not knowing what proportion of front-line workers were in fact healthcare workers, and discussion of how this may have impacted the findings.

Conclusion

- Lines 456-458 - Consider using updated terminology for monkeypox (i.e. Mpox), and also could frame this more broadly around other emerging diseases.

Minor points –

Typically we do not capitalize the ‘w’ in white, unless otherwise instructed in journal stylebook.

Line 291 – typo “we not included”

Reviewer #2: This manuscript describes the authors' experience with developing a surveillance and rapid response system to address COVID-19 misinformation in St. Louis,MI, community during the COVID pandemic.

The following issues have been identified following the review;

1. The ethical issues related to the study area are not adequately discussed. Specifically, there is a need for the authors to comment on the consent processes followed during the study.

2. The program iHeard STL from which the data emanate, has not been described adequately. There is need to provide more details regarding iHeard STL. e.g URL reference..what happened to it after the study? How is it maintained/operated?

3.There is need to comment on the sample size used for the analysis. How was this estimated? How was the sampling done (other than intentionally recruiting from the study area and targeting specified demographic characteristics).

4.Table 3 does not show all columns. The columns on the right side are not complete.

5.The present study limited analyses to univariable analysis and did not adjust for confounding factors unlike the studies used for comparison in the discussions. Unlike the present study, these studies adjusted for confounders. It is not clear why the authors did not conduct multivariable regression analyses to account for confounders. This is major limitation that limits the validity of the conclusion from this analysis.

6. Although the finding on VaxDanger by race was statistically significant, it also has a very wide confidence interval. The authors need to expand the discussion relative to this finding.

7. For the study, participants were incentivized to complete the survey. What measures are in place to ensure the high level of engagement reported in the study is maintained when participants are no longer incentivized?

6. PLOS authors have the option to publish the peer review history of their article (what does this mean?). If published, this will include your full peer review and any attached files.

Reviewer #1: No

Reviewer #2: No

---

## [Author Response · Author response to Decision Letter 0]

20 Sep 2023

Reviewer #1: General comments - This paper addresses an important topic of misinformation during pandemic times, when members of the public were exposed to high volumes of accurate information, misinformation, and disinformation. The study employed a public health surveillance model to assess prevalence of misinformation, and had high survey response rates. The study generated very interesting findings including the feasibility of a local misinformation surveillance system, and the persistence of misinformation beliefs over time despite routine counter-messaging.

1. Introduction

- Consider including a definition of “misinformation.”

Author response: Thank you for this comment. We have added a definition in the first sentence of the introduction on line 48 of the tracked changes document.

2. Consider including more information around COVID disparities to set the stage for why you specifically recruited a large proportion of Black community members to be a part of the panel.

Author response: Thank you for the comment. We added a phrase to the introduction on lines 52 and 53 of the tracked changes document that sets the stage for our Black community focus.

Methods

3. Panel composition. The authors describe their efforts to recruit a diverse panel with intentionality to include members of certain under-represented demographics including Black community members and community members from certain zip codes. They also sought representation of front-line workers in the sample and provide a definition they used for this term. It is worth noting that front-line workers encompassed healthcare workers – are these healthcare workers representative of the general population when it comes to receiving health information? Among the front-line workers, do the authors know what proportion were in healthcare?

Author response: Our operational definition of frontline workers was comprised of individuals who engage with community members through in-person, online, or phone interactions on our recruitment survey. Individuals self-classified themselves according to this definition on the recruitment survey. This inclusive approach was intended to encompass a diverse range of professionals, such as healthcare workers, phone operators, and social service personnel. We appreciate your insightful feedback and have conducted additional data analyses in response, where two reviewers coded open-ended responses on occupations as healthcare workers or not. In our analyses, we took into account missing data for 6 of the 214 panelists on occupation and two conflicts where one reviewer was uncertain and the other thought they were non-healthcare workers. To accommodate these uncertainties, we estimated the percentage of healthcare workers under four different possible scenarios:

1. The two uncertain cases and all with missing data were non-healthcare workers, 

2. The two uncertain cases were healthcare workers and all with missing data were non-healthcare workers,

3. The two uncertain cases were non-healthcare workers and all with missing data were healthcare workers,

4. The two uncertain cases and all with missing data were healthcare workers. 

Our analysis showed that healthcare workers constituted an estimated 8.4 to 12.2% of our overall participant pool, and an estimated 15.3 to 22% of these individuals fell within the category of frontline workers. We have integrated this observation into our Discussion section to provide further transparency about our sample composition. Specifically, we added the text on lines 546 to 550 on the tracked changes version of the discussion section: “In addition, an estimated 8.4 to 12.2% of study population were employed in the healthcare sector (of which an estimated 15.3 to 22% were frontline workers) (data not shown) vs. 14% of the U.S. population employed in the healthcare sector [40]. This may cause underestimated exposure rates on surveyed health-related claims.”

4. The authors should include information in the methods regarding timing of their surveys in relation to the pandemic. Was this conducted at a time of surge or low virus circulation? What were local public policies at the time? (I believe their surveys started around the time or shortly after the Delta surge in the summer of 2021).

Author response: We added two sentences to the data collection paragraph on lines 136 to 139 of the tracked changes version to describe virus circulation during this time as well as local public policies on masking. 

5. The authors provide a thorough description of how beliefs and exposures were captured.

Author response: We thank the reviewer for this comment.

6. Were surveys provided in English only? If so, would this help explain the low participation of Hispanic community members? Language of surveys should be noted in methods, and potential biases should be noted in limitations.

Author response: We thank the reviewer for this comment. We have added this information to the data collection section and to the last sentence of the limitations section of the discussion section on lines 562 to 564 of the tracked changes version.

Discussion

7. For the paragraph starting in line 383, nuanced framing of discussion in differences between racial/ethnic groups is warranted. This is especially important given stark inequities in COVID-related health outcomes between these groups. Readers should not walk away from this paragraph thinking that members of the Black community are more “gullible” (ie likely to believe false information) due to their “culture.” This is somewhat implied as the paragraph currently stands, though I do not think this was the authors’ intent. This paragraph must include a more robust discussion, including explicit consideration of how structural factors including structural violence, institutional racism, and even interpersonal racism influence trust in biomedical health system and where community members access health information. It is very important to specifically call out racism and its impact on mistrust of the biomedical system. See for example: https://www.ncbi.nlm.nih.gov/pmc/articles/PMC7241063/

Author response: We thank the reviewer for this important comment. We have made the following edits to this paragraph. We have dropped the sentence: “These findings are consistent with other recent studies that have highlighted how history, culture, and other socioecological factors influence belief of misinformation among different sociodemographic groups [33–35]” and replaced it with on lines 479 to 482 of the tracked changes version “There is a long history of African Americans being targeted with health harming products and information as well as mistrust of health care and health research among Africans Americans based on past abuses [30,31].” Further, we have replaced “higher odds of believing” with “higher odds of believing an item could be true or being uncertain whether it’s true or untrue” in the same paragraph.

8. (Similarly, the framing of racial differences in the abstract must also be addressed. The current wording regarding differences in belief of misinformation between Black and white community members is not nuanced enough and does not adequately address the systemic factors. )

Author response: We thank the reviewer for this comment. We have added a sentence at the end of the conclusion to highlight that further studies are needed to understand the observed racial differences in exposure and belief on lines 34 to 36 of the tracked changes version.

9. The exclusion of members from certain demographics in the analysis due to small numbers may have biased the data. This should be noted in the limitations.

Author response: We thank the reviewer for this comment. We added a sentence about this to the limitations section on lines 554 to 560 of the tracked changes version.

10. The limitations can also address not knowing what proportion of front-line workers were in fact healthcare workers, and discussion of how this may have impacted the findings.

Author response: We have addressed this in the discussion section as described in response 3.

Conclusion

11. Lines 456-458 - Consider using updated terminology for monkeypox (i.e. Mpox), and also could frame this more broadly around other emerging diseases.

Author response: We have changed the conclusion where noted on line 573 of the tracked changes version to include “(e.g., Mpox and other emerging diseases)”.

Minor points –

12. Typically we do not capitalize the ‘w’ in white, unless otherwise instructed in journal stylebook.

Line 291 – typo “we not included”

Author response: We thank the reviewer for this comment. We have made both of the changes as suggested. 

Reviewer #2: This manuscript describes the authors' experience with developing a surveillance and rapid response system to address COVID-19 misinformation in St. Louis, MI, community during the COVID pandemic.

The following issues have been identified following the review;

1. The ethical issues related to the study area are not adequately discussed. Specifically, there is a need for the authors to comment on the consent processes followed during the study.

Author response: We have elaborated on the process in more detail on lines 118 to 128 of the panel member subsection in the methods in the tracked changes version. Specifically, we added the following text: “We shared recruitment materials with our partners and/or distributed them while at community-based events hosted by our partners. Individuals could access the recruitment form through scanning a QR code on the recruitment flyer or email the study email address that was provided on the recruitment material. The recruitment form includes eligibility questions, followed by a full description of the project and what is asked of participants if they volunteer to participate. For those who wish to participate after reading the informed consent information, they provide contact information for payment purposes and to receive the longitudinal survey.”

2. The program iHeard STL from which the data emanate, has not been described adequately. There is need to provide more details regarding iHeard STL. e.g URL reference..what happened to it after the study? How is it maintained/operated?

Author response: In response to the reviewer’s comment, we have updated the background to provide more information on iHeard STL on lines 86 to 98 of the methods section in the tracked changes version.

3. There is need to comment on the sample size used for the analysis. How was this estimated? How was the sampling done (other than intentionally recruiting from the study area and targeting specified demographic characteristics).

Author response: We primarily recruited through the partners listed in the methods who communicated about our study to members in their networks. Importantly, iHeard STL emerged as a local response to misinformation, not a planned research project. As with many aspects of the public health response to COVID-19, our panel size was determined by urgency and practicality – how many panel members could we enlist quickly, and how many could we afford to pay? It was not estimated for scientific purposes. We have incorporated the additional information regarding the context for its origin and sample size on lines 93 to 98 of the Methods background subsection in the tracked changes version.

4. Table 3 does not show all columns. The columns on the right side are not complete.

Author response: We apologize for not catching this during submission. This has now been corrected.

5. The present study limited analyses to univariable analysis and did not adjust for confounding factors unlike the studies used for comparison in the discussions. Unlike the present study, these studies adjusted for confounders. It is not clear why the authors did not conduct multivariable regression analyses to account for confounders. This is major limitation that limits the validity of the conclusion from this analysis.

Author response: We thank the reviewer for this comment. Our primary reason for not running a multivariable regression model was the limited sample size. We now acknowledge this sample size limitation in the strengths and limitations section on lines 562 to 562 in the tracked changes version.

6. Although the finding on VaxDanger by race was statistically significant, it also has a very wide confidence interval. The authors need to expand the discussion relative to this finding.

Author response: We thank the reviewer for this comment. The wide confidence interval is because of the small sample size for this analysis. We mention this in the limitation section and ask the readers to interpret it with caution. In addition, we performed Fisher’s exact test on each of the first-time exposure and first-time belief results, and the significance remained the same. 

7. For the study, participants were incentivized to complete the survey. What measures are in place to ensure the high level of engagement reported in the study is maintained when participants are no longer incentivized?

Author response: We plan to continue to provide incentives for as long as there is funding for this surveillance system.

---

## [Decision Letter · Decision Letter 1]

10 Oct 2023

*iHeard* STL: Development and first year findings from a local surveillance and rapid response system for addressing COVID-19 and other health misinformation**

*PONE-D-22-32748R1*

*Dear Dr. Kreuter,*

*We’re pleased to inform you that your manuscript has been judged scientifically suitable for publication and will be formally accepted for publication once it meets all outstanding technical requirements.*

*Within one week, you’ll receive an e-mail detailing the required amendments. When these have been addressed, you’ll receive a formal acceptance letter and your manuscript will be scheduled for publication.*

*An invoice for payment will follow shortly after the formal acceptance. To ensure an efficient process, please log into Editorial Manager at http://www.editorialmanager.com/pone/, click the 'Update My Information' link at the top of the page, and double check that your user information is up-to-date. If you have any billing related questions, please contact our Author Billing department directly at authorbilling@plos.org.*

*If your institution or institutions have a press office, please notify them about your upcoming paper to help maximize its impact. If they’ll be preparing press materials, please inform our press team as soon as possible -- no later than 48 hours after receiving the formal acceptance. Your manuscript will remain under strict press embargo until 2 pm Eastern Time on the date of publication. For more information, please contact onepress@plos.org.*

*Kind regards,*

*Jerome Nyhalah Dinga, PhD*

Academic Editor

*PLOS ONE*

* *

*Additional Editor Comments (optional):*

* *

*Reviewers' comments:*

*Reviewer's Responses to Questions*

*

**Comments to the Author**
*

*1. If the authors have adequately addressed your comments raised in a previous round of review and you feel that this manuscript is now acceptable for publication, you may indicate that here to bypass the “Comments to the Author” section, enter your conflict of interest statement in the “Confidential to Editor” section, and submit your "Accept" recommendation.*

*Reviewer #2: All comments have been addressed*

*2. Is the manuscript technically sound, and do the data support the conclusions?*

*The manuscript must describe a technically sound piece of scientific research with data that supports the conclusions. Experiments must have been conducted rigorously, with appropriate controls, replication, and sample sizes. The conclusions must be drawn appropriately based on the data presented. *

*Reviewer #2: Yes*

*3. Has the statistical analysis been performed appropriately and rigorously? *

*Reviewer #2: Yes*

*4. Have the authors made all data underlying the findings in their manuscript fully available?*

*The PLOS Data policy requires authors to make all data underlying the findings described in their manuscript fully available without restriction, with rare exception (please refer to the Data Availability Statement in the manuscript PDF file). The data should be provided as part of the manuscript or its supporting information, or deposited to a public repository. For example, in addition to summary statistics, the data points behind means, medians and variance measures should be available. If there are restrictions on publicly sharing data—e.g. participant privacy or use of data from a third party—those must be specified.*

*Reviewer #2: Yes*

*5. Is the manuscript presented in an intelligible fashion and written in standard English?*

*PLOS ONE does not copyedit accepted manuscripts, so the language in submitted articles must be clear, correct, and unambiguous. Any typographical or grammatical errors should be corrected at revision, so please note any specific errors here.*

*Reviewer #2: Yes*

*6. Review Comments to the Author*

*Please use the space provided to explain your answers to the questions above. You may also include additional comments for the author, including concerns about dual publication, research ethics, or publication ethics. (Please upload your review as an attachment if it exceeds 20,000 characters)*

*Reviewer #2: (No Response)*

*7. PLOS authors have the option to publish the peer review history of their article (what does this mean?). If published, this will include your full peer review and any attached files.*

**

*Reviewer #2: No*

---

## [Editor Report · Acceptance letter]

26 Oct 2023

PONE-D-22-32748R1 

*iHeard* STL: Development and first year findings from a local surveillance and rapid response system for addressing COVID-19 and other health misinformation 

Dear Dr. Kreuter:

I'm pleased to inform you that your manuscript has been deemed suitable for publication in PLOS ONE. Congratulations! Your manuscript is now with our production department. 

Kind regards, 

on behalf of

Dr. Jerome Nyhalah Dinga 

Academic Editor

PLOS ONE